# Selenium Status and Supplementation Effects in Pregnancy—A Study on Mother–Child Pairs from a Single-Center Cohort

**DOI:** 10.3390/nu14153082

**Published:** 2022-07-27

**Authors:** Dorota Filipowicz, Ewelina Szczepanek-Parulska, Małgorzata Kłobus, Krzysztof Szymanowski, Thilo Samson Chillon, Sabrina Asaad, Qian Sun, Aniceta A. Mikulska-Sauermann, Marta Karaźniewicz-Łada, Franciszek K. Główka, Dominika Wietrzyk, Lutz Schomburg, Marek Ruchała

**Affiliations:** 1Department of Endocrinology, Metabolism and Internal Medicine, Poznan University of Medical Sciences, Przybyszewskiego 49, 60-355 Poznan, Poland; ewelina@ump.edu.pl (E.S.-P.); dominikawietrzyk@gmail.com (D.W.); mruchala@ump.edu.pl (M.R.); 2Department of Practical Training in Obstetrics, Poznan University of Medical Sciences, Jackowskiego 41, 60-512 Poznan, Poland; m.klobus@poczta.fm; 3Department of Perinatology and Gynaecology, Poznan University of Medical Sciences, Polna 33, 60-535 Poznan, Poland; kp.szymanowski@wp.pl; 4Institute for Experimental Endocrinology, Charité—Universitätsmedizin Berlin, D-10115 Berlin, Germany; thilo.chillon@charite.de (T.S.C.); sabrina.asaad@charite.de (S.A.); qian.sun@charite.de (Q.S.); lutz.schomburg@charite.de (L.S.); 5Department of Physical Pharmacy and Pharmacokinetics, Poznan University of Medical Sciences, Rokietnicka 3, 60-806 Poznan, Poland; amikulska@ump.edu.pl (A.A.M.-S.); mkaraz@ump.edu.pl (M.K.-Ł.); glowka@ump.edu.pl (F.K.G.)

**Keywords:** micronutrient, dietary supplement, nutrition in pregnancy, autoimmune thyroiditis, Hashimoto’s disease, hypothyroidism, cord blood, iodine, selenoprotein P, glutathione peroxidase

## Abstract

The demand for selenium (Se) increases during pregnancy since this element supports child growth, proper neuronal development and maternal thyroid function. The issue is particularly relevant for populations living in areas with a limited selenium supply, where many pregnant women opt for Se supplementation. The efficiency of this measure is unknown, although it seems vital in the prevention of severe Se deficiency. In order to evaluate this hypothesis, an observational study was conducted in Poland, where Se deficiency is prevalent. Pregnant women were invited to participate in the study and provided serum samples at the end of pregnancy (*n* = 115). Information on the supplemental intake of micronutrients was recorded in a face-to-face interview. In addition, serum samples were isolated from the cord blood of newborns at delivery (*n* = 112) and included in the analyses. Thyroid hormone status was evaluated by routine laboratory tests, and Se status was determined by total Se and selenoprotein P (SELENOP) concentrations and extracellular glutathione peroxidase (GPX3) activity. The three parameters of Se status correlated strongly within the group of mothers and within the group of newborns, with an additional significant correlation found among mother–child pairs. One-third of mothers reported additional Se intake, mainly as a component of multi-micronutrient supplements, at a mean (±SD) dosage of 42 ± 14 µg Se/day. Despite this regime, most of the women presented an insufficient Se status, with 79% of mothers displaying serum Se concentrations below 70 µg/L (indicating Se deficiency) and 22% showing levels below 45.9 µg/L (severe Se deficiency). The inadequate Se supply was also reflected in relatively low SELENOP concentrations and GPX3 activity. Neither total Se nor SELENOP or GPX3 levels were significantly higher in the group of mothers reporting the intake of supplements than in the non-supplementing group. Nevertheless, elevated SELENOP concentrations were observed in the subgroup receiving supplements with more than 55 µg/day. We conclude that the self-administered supplementation of small Se dosages was not sufficient to achieve replete Se status in the micronutrient scant area. However, the maternal Se deficit measured by either Se, SELENOP or GPX3 was transferred from mothers to the newborns, as the parameters correlated strongly in the mother–newborn pairs of samples. It is vital to re-evaluate the guidelines concerning pregnancy care and monitoring of micronutrient status during pregnancy, in particular in areas where deficiencies are present.

## 1. Introduction

Pregnancy is a period of increased requirement for micronutrients, which are essential for fetal growth and neurogenesis. Selenium (Se) is a vital trace element needed for the biosynthesis of selenoprotein. Selenoproteins with a prominent function in pregnancy and control of thyroid hormone signaling and nervous system development include the iodothyronine deiodinases (DIO), controlling activation and inactivation of thyroid hormone, the glutathione peroxidases (GPX3) and thioredoxin reductases (TXNRD), controlling redox-sensitive and protective pathways, and other specific selenoproteins involved in quality control, structural integrity and metabolic signaling [1,2]. Se deficiency is associated with an increased risk of gestational hypertension, gestational diabetes, and pregnancy complications (preterm birth, miscarriages, low birth weight) as well as poor neurodevelopment of newborns [3]. In addition, preventing Se deficiency in pregnancy reduces the risk of postpartum thyroiditis and depression [4]. Together with Se, iodine (I) accounts for neuronal migration and myelination during fetal nervous system formation, lack of which results in impaired neurodevelopment and cretinism [5,6,7,8]. 

Both trace elements play a central role in the pathogenesis of hypothyroidism (HT) and autoimmune thyroiditis (AIT) [9]. Moreover, pregnancy is a crucial time for female health and constitutes a potential triggering factor for thyroid diseases, where HT and AIT are particularly prevalent during gestation in iodine-deficient areas [10]. According to recent reports, pregnant and breastfeeding women in Poland are at high risk for both I and Se deficiency [11,12]. A systematic I deficiency prevention program was introduced in Poland in 1997, and the necessity of I supplementation in the course of pregnancy has been highlighted in endocrine practice guidelines [13]. Nevertheless, an increased requirement for Se during pregnancy is less well known and appreciated. A sufficient Se supply in pregnancy is neither mentioned in the European nor in the latest Polish recommendations, whereas the American Thyroid Association (ATA) actively discourages supplemental Se intake due to the lack of sufficient data and in view of a better general Se supply in North America [14,15,16]. Despite the scarcity of official recommendations, Se is commonly prescribed or self-administered and often taken as a component of multi-micronutrient supplements in clinical practice [17]. A large body of preclinical and clinical evidence supports the significance of preventing Se deficiency in pregnancy, in particular for women with thyroid hormone-related diseases and when residing in an area with a limited Se supply [18,19]. 

Given its importance during gestation [20,21], there is a need to improve understanding and monitoring of trace element status in pregnant mothers and newborns in a real-life setting. In order to contribute to this issue, our study was conducted in Poland with known Se deficiency and designed to characterize Se status in both mothers and newborns, with a particular emphasis on taking Se-containing supplements and how Se deficiency affects the thyroid hormone axis. 

## 2. Materials and Methods

### 2.1. Materials

This was an observational study conducted between 2019 and 2021. One hundred fifteen Caucasian, healthy women with HT and/or AIT were included in the research prior to delivery at the obstetric department of Gynecological and Obstetric Clinical Hospital of Poznan University of Medical Sciences (tertiary reference center). Serum thyrotropin (TSH), free tetraiodothyronine (fT4), free triiodothyronine (fT3) and autoantibodies to TSH-receptor (TRAb), thyreoperoxidase (a-TPO) and thyreoglobulin (a-Tg) were determined routinely in the laboratory. The results were used to distinguish three groups of women: group 1–healthy (TSH 0.27–2.5 mIU/L, no a-TPO or a-Tg, no levothyroxine (LT4) use), group 2–HT (TSH > 2.5 mIU/L, or normal/decreased on LT4 therapy, including patients with AIT and without AIT), and group 3–AIT (positive a-TPO or/and a-Tg, including those with TSH > 2.5 mIU/L and normal/decreased on LT4 therapy). One subject was later excluded due to the lack of material resulting from perinatal complications. Accordingly, the material from 114 mothers was finally analyzed. In addition, samples from 112 newborns were available for analysis, increasing the full cohort size to 226 samples for Se analysis (including 9 twin pregnancies). Two mothers were not assigned to the three thyroid-related groups due to the insufficient material and a lack of thyroid hormone panel assessment and were removed from the analyses involving the thyroid status. The mother–child pair analyses encompassed 101 pairs, after exclusion of pairs with lacking newborn results due to insufficient cord blood volume. Other exclusion criteria in the study group were the history of any malignancy, thyroid carcinoma or severe kidney or liver diseases (except for benign lesions, such as renal or liver cysts), and in the group of healthy subjects, a prevalent thyroid disease (except for thyroid nodules). Additionally, patients traveling abroad for a period longer than four weeks (the vast majority were residents of western Poland) or declaring any restrictive diet (fish-rich, vegetarian or vegan) were not included. 

### 2.2. Methods

On admission to the ward, a first consideration and a medical interview were performed by the qualified instructed midwives. A venous blood sample was taken from the mother and stored deep-frozen in two aliquots. During the third phase of delivery, cord blood (up to 2 mL) was squeezed from the placental neonatal vessels and stored frozen. One sample obtained from the mother and the newborn was used for thyroid status assessment, and the other for trace element evaluation. 

Concentrations of TSH, fT3, fT4, a-TPO and a-Tg were measured by commercial kits using electrochemiluminescence (ECLIA) (Hitachi and Roche Diagnostics), and the concentration of TRAb was measured using radioimmunoassay (RIA) (BRAHMS Diagnostics, Berlin, Germany) on a Cobas e601 analyzer (Indianapolis, IN, USA). Se status assessment was performed in the analytical lab of the Institute for Experimental Endocrinology, Charité-Universitätsmedizin Berlin, by scientists blinded to the clinical information. Se was measured by the total reflection X-ray fluorescence (TXRF) analysis (TXRF spectrometer S4 T-STAR; Bruker Nano GmbH, Berlin, Germany) [22], SELENOP was assessed by a validated colorimetric enzyme immunoassay (selenOtest; selenOmed GmbH, Berlin, Germany) [23], and GPX3 activity was determined using a coupled enzymatic assay with t-butyl hydroperoxide as a substrate [24]. The reference ranges for the comparison were derived from the data determined during the analysis of a subsample of the European Prospective Investigation into Cancer and Nutrition (EPIC) study cohort of healthy European adults [25]. The threshold for Se deficiency of <70 µg/L was determined on the basis of the literature and the previous findings [26].

### 2.3. Statistical Analysis

Statistical analysis was performed on TIBCO Software Inc. (2017), Statistica 13 (Statsoft, Tulsa, OK, USA), and with the software R (The R Foundation, Vienna, Austria), version 4.0.2., and RStudio, version 1.02.5042. The distribution of the variables was assessed using the Shapiro–Wilk’s test. The analyses were conducted between the groups of unrelated subjects who were classified as mothers and newborns and/or in the mother–child pairs. For comparisons between the groups, the Mann–Whitney U test and the ANOVA Kruskal–Wallis test were used. Comparisons within mother–child pairs were performed using the Wilcoxon paired samples test. Correlations between continuous variables were carried out using the Spearman R test and the Chi2 NW test (the highest reliability). Results are presented as a mean and SD or a median and interquartile range (IQR, Q1–Q3). A *p*-value < 0.05 was considered statistically significant.

## 3. Results

### 3.1. Group Characteristics

An overview of the mothers and children enrolled and included in this observational study is presented (Figure 1), with the conducted clinical and biochemical analyses.

The mean age of mothers was 34 ± 4 years, the mean body weight before pregnancy was 64.7 ± 11.7 kg, and 94% of pregnancies were term deliveries (at the mean pregnancy week 38.8 ± 1.6). The thyroid hormone profiles of the mothers and the newborns according to the analyzed categories are shown in Table 1.

### 3.2. Correlation Analysis of Se Status Biomarkers in the Mothers and the Newborns

The comparison of different biomarkers involving Se status to each other indicated strong, although not perfect, correlations (Figure 2). In the group of mothers, the total serum Se correlated positively to SELENOP (*p* < 0.001, r = 0.78), and to GPX3 activity (*p* < 0.001, r = 0.783). Although correlation of SELENOP with GPX3 was strong, yet slightly less stringent (*p* < 0.001, r = 0.644). The same applies to newborns, confirming a relatively poor Se status in both groups, insufficient to saturate selenoprotein expression. The wider range of Se status in mothers indicated a higher fraction of individuals with a severe Se deficiency in this group than in the newborns, despite the family relationship between both groups. This, in turn, indicates some regulated supply from mothers to newborns and not unregulated transfer. Although all Se status parameters are significantly different in the newborns and in the mothers, a positive correlation of the parameters within mother–newborn pairs is present: Se (z = 8.07, *p* < 0.001), GPX3 (z = 7.41, *p* < 0.001), SELENOP (z = 5.15, *p* < 0.001).

### 3.3. Selenium Status in the Mothers and the Newborns

The entire collection of serum samples was analyzed for both total Se and SELENOP concentrations and classified according to the thresholds of deficiency. Using a commonly used cut-off for serum Se of 70 µg/L, the majority of samples were classified as Se-deficient (Figure 3A,B). Choosing a more stringent threshold, a fifth of the mothers and four-fifth of the newborns were severely deficient and displayed Se concentrations below 45.9 µg/L. A similar picture was obtained by SELENOP analysis. Taking the 95%-reference range of healthy European adults as threshold, three-quarters of all samples (77%) was below the 2.5th centile, i.e., below the threshold for deficiency (Figure 3A,B). Comparing the classification according to the total serum Se deficiency and SELENOP deficiency, the results are highly congruent with few exceptions only (Figure 3C). The differences observed may be associated with the recent dietary or supplemental Se intake, which would directly affect the total serum Se concentrations, although they would not immediately impact acute circulating SELENOP levels.

A comparison between the group of mothers and the children indicates that all three biomarkers of Se status are significantly lower in newborns than in adult women (Table 2). The relative difference is similar for the total Se and GPX3 (children presenting 67% and 69% of the levels found in mothers, respectively) and less pronounced for SELENOP (children showing 74% of the maternal status). Nevertheless, all three biomarkers highlight, on average, the relatively more pronounced Se deficit in the newborns as compared to the mothers, which is potentially related to both the general deficiency of this element in Poland, as well as the immature Se metabolism in newborns.

A direct comparison between the mothers and their newborns emphasizes that the differences are not uniform but rather pair-specific (Figure 4). Notably, most mothers with Se deficiency transfer the deficit to their newborns, with only a few exceptions. Furthermore, the overall picture of a relatively low Se status of the newborns appears not to be driven by individual pairs but constitutes a general trend, albeit with notable exceptions.

### 3.4. Effects of Se Supplementation

About one-third of women (35/115) reported supplemental Se intake during pregnancy, using different supplement types and dosages. Se was most frequently administered as one component of multi-micronutrient supplements, with a mean dosage (±SD) of 42 ± 14 µg/day, with a range of 6.25–55.0 µg/day (min–max, respectively). Three (9%) of the women supplemented Se in the first and second trimester only (sodium selenate; 55, *n* = 2 or 26 µg/day), whereas the other 32 women ingested the supplement in the course of the entire pregnancy (Table 3).

Comparing the mothers and the children with regard to Se supplementation, no significant overall differences in the Se status parameters were observed between the two groups (Table 4). However, considering thresholds for Se deficiency (<70 µg/L and <45.9 µg/L), non-supplementing women were over-represented in both categories (71% and 76% of cases, respectively).

When subdividing the group of women according to the Se supplements used into groups of low (<55 µg/day) or moderate (≥55 µg/day) daily dosages of Se, a difference in the median SELENOP (IQR) was observed, with significantly higher levels in the group receiving higher daily dosages; SELENOP [mg/L]: 1.84 (1.49–2.20) vs. 3.17 (2.55–3.88) mg/L, U = 59, *p* = 0.006. Strong Se deficiency (Se < 45.9 µg/L) was more prevalent in the group of women taking a low daily dosage of Se as compared to the women receiving higher Se dosages (19 vs. 12).

### 3.5. Relationship between the Status and Supplementation of Se and the Parameters of Thyroid Diseases

Newborns deficient in Se presented higher median (IQR) TRAb concentrations than the newborns with a sufficient supplementation; 0.30 (0.30–0.56) vs. 0.27 (0.07–0.37), U = 440, *p* = 0.02. This tendency was also found in the Se deficient mothers with a borderline statistical significance; 0.63 (0.38–0.82) vs. 0.30 (0.27–0.70), U = 639.5, *p* = 0.07. The TRAb levels of newborns correlated inversely with their SELENOP concentrations (R = −0.22, *p* = 0.038), and their newborn Se level (R = −0.27, *p* = 0.011). 

## 4. Discussion

In this observational exploratory study, the Se status and potential effects of moderate Se supplementation were investigated in a cohort of pregnant women living in an area with a low habitual dietary supply of Se. Our results indicate a pronounced deficiency in this essential trace element in the majority of pregnancies, both in the mothers and newborns. Daily intake of a multi-micronutrient supplement produced only marginal effects, and the efficiency on selenoprotein expression was observed when the daily Se dosage exceeded 55 µg. Importantly, the analysis of the paired mother/newborn samples indicated that the Se deficit was transferred from mothers to children in a personalized manner, i.e., a general declining trend from mother to newborn was observed, with notable individual exceptions, indicating additional factors affecting Se supply of the newborns in the perinatal period.

According to the earlier reports regarding Se supply in healthy pregnancies in Poland over the last decades, Se concentrations have changed over time. The concentrations were significantly lower at the time of pregnancy and in newborns, with plasma Se levels of 48 μg/L in the cord blood and 55 μg/L in mothers at delivery in the early 1980s, which further decreased to 33 μg/L and 40 μg/L, respectively, in the late 1990s [27]. In fact, general plasma Se level in Polish subjects was also reported as relatively low (50–55 μg/L) and attributed to an inadequate dietary intake.

The National Institutes of Health (NIH) in the US opt for Se recommended dietary allowance (RDA) in adults to be 55 μg/day, 60 μg/day in the course of pregnancy and 70 μg/day during lactation [28]. Nevertheless, a higher daily Se demand was established by the European Food Safety Authority (EFSA), reaching 70 μg for adults (including pregnancy) and 85 μg in breastfeeding women [29]. To the best of the authors’ knowledge, no Polish recommendations concerning Se optimal intake with diet have been established. However, the German, Austrian and Swiss nutrition societies (DACH) [30] suggest body mass adjusted daily recommended intake for pregnant and non-pregnant women (60 μg), for men (70 μg) and breastfeeding females (75 μg), and recommend providing 10 μg of Se to children between 0 to under 4 months, and 15 μg between the ages 4 to 12 months. The lack of increase in demand during pregnancy was motivated by a low fetal Se requirement of 2 μg/day provided by the trans-placental transfer. However, in these epidemiological guidelines, the authors emphasized that the most reliable individual Se status marker was SELENOP saturation, which in our study was found to be highly correlated in both the mothers and the newborns, indicating a sub-optimal Se supply. This reference also assumes a proper Se maternal intake, which is known to be insufficient (in one study, 30–40 μg/day) in individuals in Poland and in Europe, including pregnant women [27]. Moreover, the provided recommendation is associated with healthy individuals, and it does not address thyroid diseases. In the current study, we emphasized the need for local recommendations which would account for personalized factors. 

A previous study from the Warsaw district indicated severe Se deficiency at the end of pregnancy, irrespective of AIT diagnosis (Se 52 μg/L, SELENOP 1.5 mg/L), or health status (Se 48 μg/L, SELENOP 1.3 mg/L), reaching severe Se deficits (<45 μg/L, according to the WHO classification) in 28.6% and 35.5% of pregnancies, respectively [12]. In the study presented here, serum Se of the studied mothers was similarly low, at 54 μg/L, and even lower in the cord blood (36 μg/L). Additionally, the concentrations of SELENOP were slightly higher, and severe deficits were found in a fraction of the mothers (20.2%) and simultaneously in the majority of newborns (80%). Serum Se of <50 μg/L was reported in 31% of pregnant females from Warsaw and in 32% of pregnancies in our study, indicating a very consistent representation of a general Se deficiency in pregnancies in Poland. 

In order to assess Se supply, we used reference ranges from a large cohort of healthy European subjects recruited into the EPIC cross-sectional study (*n* = 1915, average plasma Se 84.8 µg/L and SELENOP 4.4 mg/L) [25]. The threshold for Se deficiency was deduced from the data as 45.9 µg/L, which denoted the 2.5th percentile of the EPIC results, and was similar to a previous Polish study [25,27]. However, the deduced threshold from the EPIC cohort referred to adults, and a similarly large database for newborns and children was missing. A recent study involving newborns evaluated serum Se on day 0 in healthy subjects, reporting average Se levels of 36.2 µg/L, which corresponded to 40% of the values in adults [31]. Our analysis reported a very similar neonatal Se concentration level, which corresponded on average to 33% of the concentration of the mothers. Both results were below the level observed in a group of German newborns (about 50.6 µg/L) [32]. Our results indicating lower total serum Se concentrations and GPX3 activities in the newborns in comparison to their mothers are also in line with a similar study from another area of poor Se status, i.e., New Zealand [33]. A considerably decreased Se and SELENOP concentrations were also observed in preterm births and in neonatal infections, where SELENOP increased following antibiotic therapy due to the improved hepatic selenoprotein biosynthesis at day 3 of life [34], as well as potentially due to the disrupting effects of the antibiotic during selenoprotein translation [35,36]. Since our group comprised mostly term deliveries and no severe neonatal infection was noted, the difference between the mothers and the newborns depended rather on the micronutrient status of the mother and not on the disease-dependent parameters. 

In our study, Se status was assessed by serum Se concentration and two selenoproteins (GPX3 activity and SELENOP concentration). SELENOP is a main body Se transporter and constitutes up to 60% of total blood Se [26]. The positive and stringent linear correlation between serum Se and the two protein biomarkers reflects insufficient selenoprotein biosynthesis and the lack of saturation with Se, and thus another strong indication of an insufficient Se status of both the mothers and their children. Se supplementation has been found to efficiently increase selenoprotein biosynthesis and thereby serum Se status, and data regarding the maximal tolerable dose are available [37]. Additionally, in the course of pregnancy, supplemental Se has been reported to prevent postpartum diseases in mothers in some, although not in all studies [38,39]. After pregnancy, supplemental Se can be transferred into mothers’ milk and transferred to the newborn during lactation [40,41], mainly in the form of SELENOP, which allows targeted Se transport and tissue accumulation [41,42]. Therefore, pregnant and breastfeeding mothers are able to support their growing newborn children with the trace element, even when their own supplies are low and deficiency is present. However, this interrelationship apparently reaches its limits when the trace element is chronically in low supply [43]. Therefore, breast milk analyses would provide additional and crucial information with regard to the nutritional status of the mother and her Se deficiency, very similar to I, where the breast milk I appears to constitute the more reliable parameter than the urinary I [43]. However, our data indicate that despite the general trend, the interrelationship of the Se status between mother and newborn is of an individual personalized character, and additional parameters which need to be identified affect the transfer as well as the resulting gradient.

Up to date, active Se supplementation has been officially recommended only in the case of mild Graves’ ophthalmopathy (GO) [44]. However, a considerable number of studies provide arguments for its beneficial influence on AIT, HT and in pregnancy [3]. Nevertheless, the amount of Se necessary in the course of gestation to prevent deficiency has not been determined so far, although it has been established that it depends strongly on the area of residence and the habitual Se intake of a given population. It is unlikely that a large dosage is needed where sufficient basic Se intake is ensured, e.g., in North America or in other Se-rich areas [45]. In our group of pregnancies, more than one-third of females reported the intake of Se-containing supplements, which seems a reasonable measure to prevent severe deficiency. However, the data obtained indicate that a certain amount is needed to reach detectable effects and that the daily dosage below 55 µg of the supplemental Se has not improved the Se status according to any of the three Se biomarkers analyzed. Importantly, Se was administered mainly as inorganic salts and not in the organic form (selenomethionine), which may present higher bioavailability [18]. Supplementation with 60 μg/day of selenomethionine is recommended for healthy women and even 200 μg/day for women at risk of thyroid postpartum disturbances, although these amounts are normally not provided by a common prenatal trace element preparation [46]. Similar to our observations have been reported from a study conducted in Latvia where, despite the usage of a broad supplement (70%), pregnant females failed to achieve the optimal Se status [47]. 

### Strengths and Limitations

Our study has several strengths and notable limitations. The study cohort was of sufficient size to monitor Se status, both in the mothers and their newborns, as well as to analyze the general trends and individual patterns. The parallel quantification of three Se status biomarkers allowed for a detailed assessment of the Se status and provided a comprehensive picture of the prevalent Se deficiency in both the mothers and their newborns. In view of the high inter-parameter correlation and relatively low concentrations of any of the three biomarkers, all three parameters appear well suitable to classify the Se status of pregnant mothers in this population, as well as to detect and monitor deficiency and supplementation effects. The assessment of SELENOP seems to be slightly advantageous, as it reflected the anabolic metabolism of the supplemental Se and appeared more sensitive than total Se concentrations in detecting the supplementation effects. In view of the known health risks involving both mothers and children in severe Se deficiency, the data obtained strongly support Se substitution, i.e., support by moderate supplemental Se intake. The results from the fraction of women, who had already chosen to self-supplement and reported on their intakes, clearly highlight that a certain dosage is necessary to elicit positive effects on the Se status and SELENOP expression and that minimal dosages may not be sufficient. 

However, our study is not without limitations. Firstly, the enrolled group of women is heterogeneous in terms of thyroid-related diseases, underlying autoimmune diseases and LT4 treatment. Secondly, no detailed information on the dietary patterns was recorded, and no suitable food frequency questionnaire was used to obtain information on the relevant nutritional confounders. Thirdly, one serum sample per mother and per child only was available for the purpose of the analysis (i.e., the cord blood collection due to technical difficulties was limited), and the information on the supplemental trace element intake was acquired by a single medical interview; therefore, a risk of underreporting or incorrect information should be noted. Finally, the study design was observational, and it was impossible to deduce causal effects, even though the congruent findings of dose-dependency of supplemental intake with improved biomarkers support the quality of the self-reports and laboratory analyses. 

## 5. Conclusions

Our study adds to the body of evidence reporting a low and insufficient Se status of pregnant women in central Poland and emphasizes the direct interrelationship of Se deficit in mothers, which is directly reflected in a poor Se status in the newborns. Overall, self-reported Se supplementation by pregnant females appeared ineffective for optimal micronutrient restoration. However, the usage of moderate (>55 µg) instead of low doses improved the Se status. In view of these findings, it is necessary to introduce better nutritional education for pregnant women and to provide them with an additional supplementary intake of this element in areas with a known Se deficiency. Local guidelines regarding supplementation in pregnancy should be revised according to the available studies concerning the general Se status in the given population. Considering the significant worldwide differences in the habitual Se intake, no general guidelines will ever be drafted advocating for or against Se supplementation, as the baseline intakes demonstrate very strong regional variability, thus rendering extra requirements in pregnancy area-specific. In addition to the geographical differences, individual dietary patterns are also relevant, with chronic illness, vegetarians and particularly vegans at an increased risk for Se insufficiency. Therefore, the recommendations in pregnancy should be personalized, ideally based on a laboratory analysis of the Se status involving any of the biomarkers used in this study to correctly adjust the supplement dosage to achieve a satisfactory supply and Se sufficiency.

## Figures and Tables

**Figure 1 nutrients-14-03082-f001:**
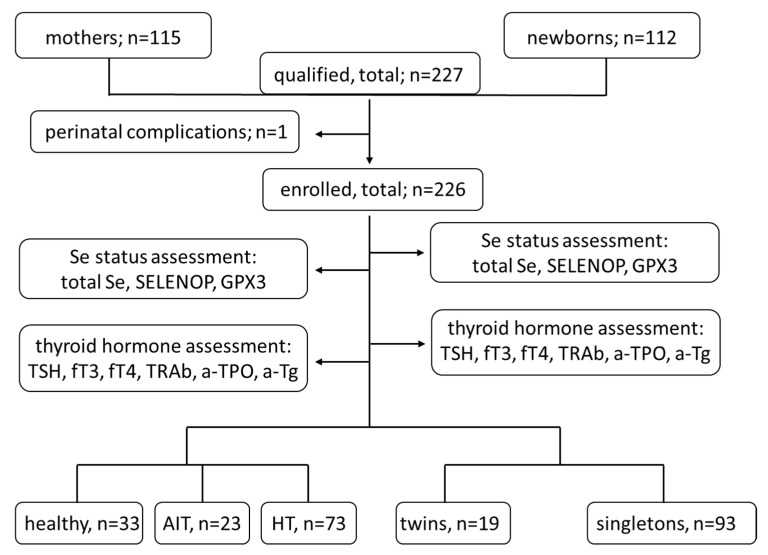
Overview of the mothers and children enrolled in the analysis. The group of mothers was finally divided into three groups of healthy women and mothers with different thyroid diseases, and the group of newborns consisted of a few twins and a majority of singletons. HT group comprised 15 patients with concomitant AIT, including euthyroid subjects on levothyroxine; one child from twin pregnancy had insufficient cord blood volume to perform the analysis. In terms of the related mothers and newborns calculations, 101 pairs were included in the analyses. AIT—autoimmune thyroiditis; HT—hypothyroidism; TSH—thyrotropin; fT4—free tetraiodothyronine; fT3—free triiodothyronine; TRAb—autoantibodies to TSH-receptor; a-TPO—autoantibodies to thyreoperoxidase; a-Tg—autoantibodies to thyroglobulin; Se—selenium; SELENOP—selenoprotein P; GPX3—plasma glutathione peroxidase.

**Figure 2 nutrients-14-03082-f002:**
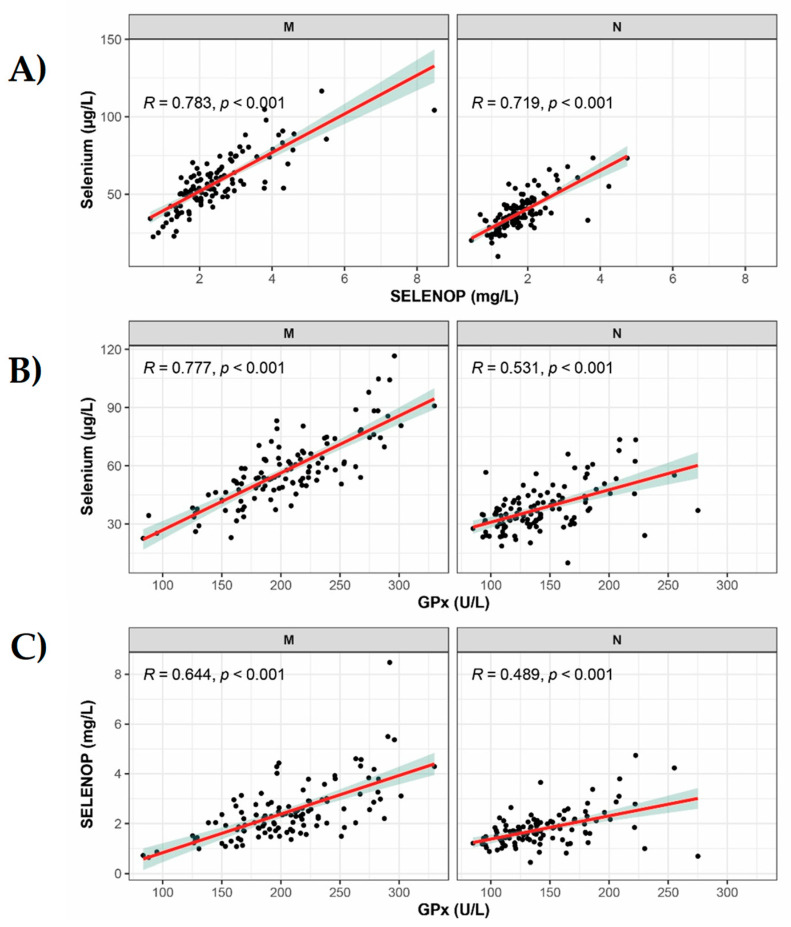
Correlation analysis of Se status biomarkers in the mothers and the newborns. Significant positive correlations were observed for all three biomarkers in all possible combinations, both in the full cohort of samples as well as in the groups of mothers (M, left panel) and newborns (N, right panel) separately. The linear positive correlation of (**A**) Se with SELENOP was most pronounced and stringent, in particular in newborns. (**B**) Serum Se and GPX3 activity correlated to a similar extent as (**C**) SELENOP and GPX3 activity. Correlation analyses conducted by Spearman’s R test; linear regression results are indicated as red lines with the 95% confidence intervals as green shadows.

**Figure 3 nutrients-14-03082-f003:**
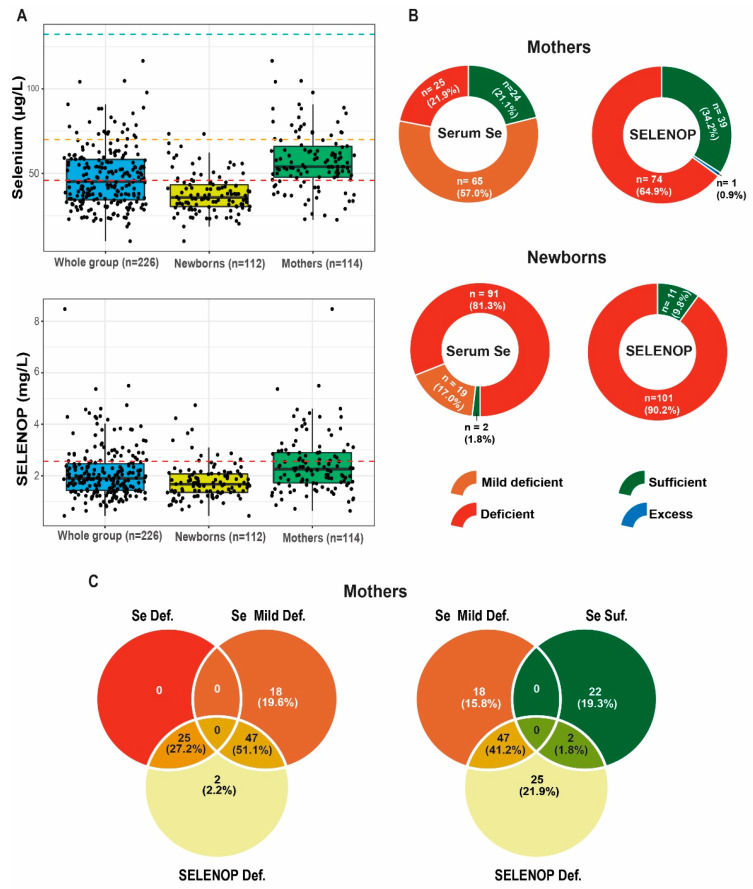
Overview of the Se status of the entire cohort. (**A**) Total serum Se and SELENOP was determined in the serum of the mothers and the newborns and categorized according to the range of EPIC-Study. Cut-off for deficiency: sufficiency; [Se] > 70 µg/L (green dashed line), mild deficiency; [Se] < 70 µg/L and >45.9 µg/L (orange dashed line) versus deficiency; [Se] < 45.9 µg/L (red dashed line). The majority of both mothers and newborns are below the commonly used threshold of 70 µg/L (mild deficiency). Cut-off for deficiency according to the SELENOP status: excess; [SELENOP] > 6.63 mg/L, sufficiency; [SELENOP] < 6.63 mg/L and >2.56 mg/L, versus deficiency; [SELENOP] < 2.56 mg/L (red dashed line). The majority of both mothers and newborns are below the threshold of 2.56 mg/L (deficiency). (**B**) Doughnut charts presenting the relative fraction of subjects with sufficient or deficient Se status according to the serum Se and SELENOP, respectively. (**C**) Venn diagrams highlighting the overlap and discordance of the simultaneous Se and SELENOP deficiency.

**Figure 4 nutrients-14-03082-f004:**
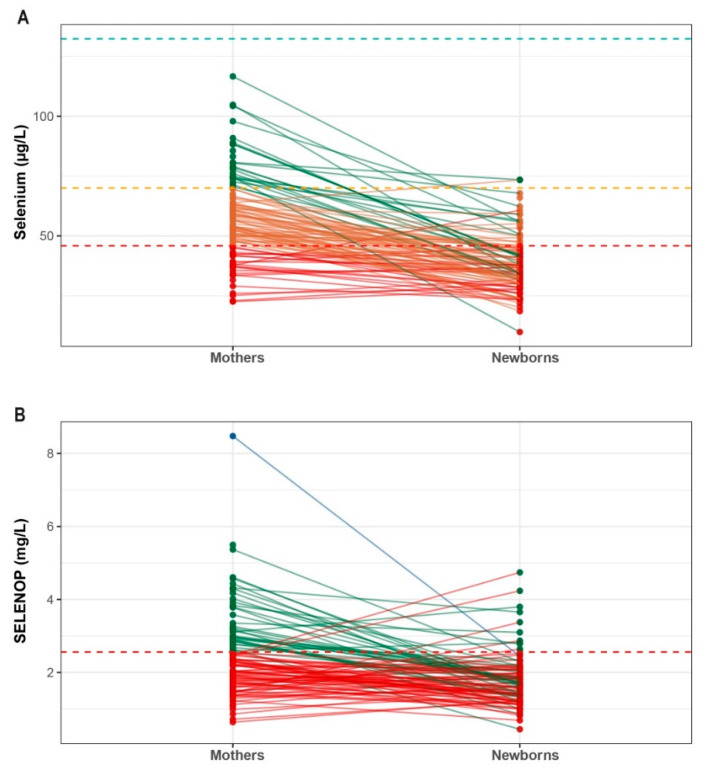
Comparison of Se and SELENOP concentrations between the mothers and their newborns. Despite the general tendency that concentration levels in newborns are lower as compared to those observed in their mothers, certain mother–infant pairs show particularly pronounced differences in concentration, suggesting that the interaction is subject to individual control mechanisms. (**A**) Color code indicates: Se deficiency (red) with [Se] < 45.9 µg/L, mild deficiency (orange) with [Se] > 45.9 µg/L and <70 µg/L, or sufficiency (green) with [Se] > 70 µg/L. (**B**) The color code used to highlight the SELENOP status includes: deficiency (red) with [SELENOP] < 2.56 mg/L, sufficiency (green) with [SELENOP] > 2.56 mg/L and <6.63 mg/L, versus excess (blue) with [SELENOP] > 6.63 mg/L.

**Table 1 nutrients-14-03082-t001:** Overview of thyroid hormone profiles in different thyroid states in mothers and children.

		Subgroup					
		Whole Group	AIT (+)	AIT (−)	TSH > 2.5 [mIU/L]	TSH 0.27–2.5 [mIU/L]	Healthy Group
		Me [Q1–Q3]	Me [Q1–Q3]	Me [Q1–Q3]	Me [Q1–Q3]	Me [Q1–Q3]	Me [Q1–Q3]
**Mother**	**TSH** **[mIU/L]**	2.09 [1.31 –2.86]	1.65 [1.23–2.46]	2.15 [1.31–2.93]	3.21 * [2.93–4.34]	1.56 * [1.11–2.1]	1.79 [1.17–2.19]
	**ft3** ** [pmol/L]**	4.03 [3.62–4.38]	3.71 ** [3.39–4.14]	4.17 ** [3.73–4.43]	3.96 *** [3.44–4.36]	4.05 [3.71–4.42]	4.33 *** [3.92–4.59]
	**ft4** ** [pmol/L]**	13.6 [12.2–15.4]	13.8 [12.2–16.3]	13.6 [12.2–15.1]	13.7 [12.6–16.3]	13.5 [12.1–14.8]	13.4 [11.4–14.5]
	**a-TPO** ** [IU/mL]**	12 [10–20]	82 * [46–173]	11 * [10–14]	12 [10–20]	12 [10–17]	11 [10–13]
	**a-Tg** ** [IU/mL]**	14 [11–19]	38 * [22–104]	13 * [11–15]	14 [10–17]	14 [11–19]	13 [11–15]
	**TRAb** ** [IU/L]**	0.46 [0.3–0.71]	0.65 [0.3–1.08]	0.35 [0.3–0.68]	0.48 [0.3–0.68]	0.33 [0.29–0.87]	0.47 [0.27–0.86]
**Child**	**TSH** ** [mIU/L]**	7.87 [5.77–11.15]	6.59 [5.37–9.89]	8.44 [5.85–11.55]	9.22 *** [7.25–11]	7.08 [5.49–10.8]	6.43 *** [4.77–9.1]
	**ft3** ** [pmol/L]**	2.07 [1.8–2.37]	2.02 [1.65–2.22]	2.07 [1.83–2.43]	2.1 [1.8–2.42]	2.01 [1.75–2.35]	2.07 [1.83–2.85]
	**ft4** ** [pmol/L]**	16.4 [15.1–17.6]	16.45 [14.7–17.4]	16.4 [15.2–17.9]	16.4 [15–17.9]	16.3 [15.1–17.4]	16.2 [15.1–17.9]
	**a-TPO** ** [IU/mL]**	9 [9–11]	60.5 * [28–105]	9 * [9–9]	9 [9–9]	9 [9–13]	9 [9–9]
	**a-Tg** ** [IU/mL]**	13 [10–14]	18 * [14–47]	12 * [10–14]	13 [10–14]	13 [10–16]	12 [10–14]
	**TRAb** ** [IU/L]**	0.3 [0.23–0.43]	0.4 [0.3–0.9]	0.3 [0.23–0.43]	0.3 [0.23–0.41]	0.3 [0.26–0.53]	0.3 [0.12–0.57]

AIT (+)—autoimmune thyroiditis; AIT (−)—no autoimmune thyroiditis; Me—median; Q1—first quartile; Q3—third quartile; TSH—thyrotropin; ft3—free triiodothyronine; ft4—free tetraiodothyronine; TRAb—autoantibodies to TSH-receptor; a-TPO—autoantibodies to thyreoperoxidase; a-Tg—autoantibodies to thyroglobulin. * *p* < 0.001, ** *p* = 0.01, and *** *p* = 0.02 (Mann–Whitney U test).

**Table 2 nutrients-14-03082-t002:** Selenium status in the mothers and the newborns.

	Group		U	*p **	*T*	*p ***
	Mother	Child				
	Me [Q1–Q3]	Me [Q1–Q3]				
Se [µg/L]	54 [48–66]	36 [30–43]	2113	<0.001	226	<0.001
GPx [U/L]	199 [167–232]	137 [112–164]	1926	<0.001	426.5	<0.001
SELENOP [mg/L]	2.3 [1.7–2.9]	1.7 [1.4–2.1]	3647	<0.001	1114	<0.001

Se—selenium; GPX3—glutathione peroxidase activity; SELENOP—selenoprotein P; Me—median; Q1—first quartile; Q3—third quartile; U—the result of Mann–Whitney U test; *p* *—level of significance for Mann–Whitney U test; *T*—the result of Wilcoxon test; *p* **—level of significance for Wilcoxon test.

**Table 3 nutrients-14-03082-t003:** Self-reported intake of Se-containing supplements during pregnancy.

Percentage of Women on Various Selenium Supplements (Number from Total *n* = 35)	Formula	Dose [µg/per Day]
**51% (20)**	sodium selenate	55
**40% (14)**	sodium selenate	26/30
**3% (1)**	L-selenomethionine	6.25

**Table 4 nutrients-14-03082-t004:** Impact of supplementation on Se status in the mothers and the children.

	Se Supplementation		U	*p*
	No	Yes		
	Me [Q1–Q3]	Me [Q1–Q3]		
**Se M [µg/L]**	54 [46–64]	58 [50–71]	1040	0.42
**GPX3 M [U/L]**	199 [170–225]	208.5 [179–254]	1049	0.46
**SELENOP M [mg/L]**	2.2 [1.7–2.6]	2.3 [1.7–3.6]	986	0.23
**Se C [µg/L]**	35 [30–43]	37 [32–45]	950.5	0.15
**GPX3 C [U/L]**	137 [113–161]	139 [117–180]	1047	0.45
**SELENOP C [mg/L]**	1.6 [1.3–2]	1.7 [1.4–2.2]	986	0.23

M—mothers; C—children; Se—selenium; GPX3—glutathione peroxidase activity; SELENOP—selenoprotein P; Me—median; Q1—first quartile; Q3—third quartile; U—the result of Mann–Whitney U test; *p*—level of significance for Mann–Whitney U test.

## Data Availability

The data are accessible upon reasonable request to the senior author.

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
