# Peer review of "Selenium Status and Supplementation Effects in Pregnancy—A Study on Mother–Child Pairs from a Single-Center Cohort"

_nutrients, 2022, doi:10.3390/nu14153082_

Round 1

Reviewer 1 Report

This observational study conducted in Poland measured Se supplementation in pregnant mothers and Se status in mothers and infants.

Overall the manuscript if good, clearly written and easy to follow. The largest benefit with the paper is that Se status is measured in different ways both in the mothers and the infants.  It is also good that the concentration of Se in the supplements are measured.

Comments:

1) Consider changing the order in the introduction so that it starts with selenium (and not with iodine) which is what is the scope of this paper.

2) Table 1 is missing.

3) I don't understand how the thyroid hormone profiles are used when Selenium is analyzed. I don't find any stratification for hormone profile? Neither is hormone profiles used as confounders? Does the correlation between supplementation and status differ depending on hormone profile?

4) Figure 5 is of low quality.

5) How do you explain the fact that "any supplementation" does NOT have an effect on Se status, but "high supplementation" was associated with more SelenoP than "low supplementation"?

6) I miss a part in the discussion where you discuss the dietary intake of selenium in relation to your results. Do women that take supplements differ regarding selenium intake from the diet? What does the literature say? And how is that related to your results?

Reviewer 2 Report

Comments to the Authors of manuscript number: animals-1779883  entitled “Selenium status and supplementation effects in pregnancy - study on mother-child pairs from a single-centre cohort”.

The study involves mothers and their infants. The trace mineral selenium is an essential nutrient that is fundamental to human biology. As a component of selenoproteins, selenium is involved in many metabolic pathways that are critical for life. Selenium plays a key role in oxidation/reduction (redox) reactions. It is involved with reduction of hydrogen peroxide as well as decomposition of lipid and phospholipid peroxides to harmless products. Selenium deficiency is associated with increased risk of mortality, poor immune function and mood disorders. Selenium is also involved with thyroid hormone metabolism and its deficiency may affect thyroid function.

In many European countries, dietary intake of selenium is below the daily requirements. There is an ongoing heated debate concerning the need for supplementation, its forms and dosing. The value of 70-150 µg / L is considered a normal concentration of Se in the blood, Polish observations reveled its mean value at 63.5 µg / L (2016).

In general, the paper is prepared and presented in bad manner. It should be substantially corrected.

1. Abstract should be shortened. It is too much descriptive.

2. L 66 – the title informs that the paper is about Se

3. L 70 – Authors have to focus on Se

4. L 79 – this sentence is not understood. “a large body’?

5. The introduction should be rephrased according to the title or Authors have to fit the whole paper to the introduction.

6. L 91 Did Authors observe the moment of the birth?

7. L 104-105- this sentence does not have the sense. What do Authors mean?

8. L 105 – how available? what do the Authors think?

9. L 110- what is a comlete material?

10. L 111-116 If the whole number of women was 115, and finally in the study the number of women was 112, to which number this description relates? Over 115 participated in the study?

11. L 117-125 – it is too descriptive and involves too many not important data like “centrifuged, sampled, frozen..”

12. In general Material and methods should be rephrased and prepared in manner which allow to easy follow the study design.

13. figure 1. The most importand information is that related the number of pair not samples.

14. Where is the table 1?

15. Results: all the data should be presented, not only the correlation. Where are the results of the Se detection?

16. Why the table 1 and 2 should be connected

17. L 204 – Why the cut-off was 70 µg/L? it is not explained.

18. Due to the differences in the occurrence of Se in particular regions of the world, different values of its daily supply are recommended. For example, the US National Institutes of Health (NIH) recommends to adults with Se supplementation in the amount of 55 µg daily, while the recently published guidelines of the German Nutrition Society - DGE) are recommended (Germany, Austria,

Switzerland) the daily supply of Se in the amount of 70 µg in men and 60 µg in women. In Poland, the average demand for adults was estimated at 45 µg of selenium per 24 hours, but there are no objective studies confirming the validity of this assumption. Why it was not taken into consideration during the preparation of the study design?

19. L 321- reference should be added

20. L 427- “one serum sample per mother and per child only was available for analysis” what does it mean?

21. Conclusion too long

22. The paper does not correlated with the title. The introduction and performed analysis focuses on the Se and thyroid hormones but, as Authors wrote Se participate also in immulogical processes

23. The demand for Se in a pregnant and lactating woman should take into account the satisfaction of the child's needs, which in utero is about 2 µg per day, in an infant up to 4 months of age - about 10 µg per day, and between 4 and 12 months of life - 15 µg per day. It was not discussed, it is omitted.

24. with the recommended daily supply of 60 µg for adult women, there is no need to increase the consumption of Se during pregnancy, while nursing women should take Se in the amount of 75 µg daily. It is not mentioned

25. Se participate in many life aspects: oxidative stress, diseases of cardiovascular system, cancers, virus infections, immunological system, reproduction, psychological functions, thyroid gland diseases (the organ with the highest concentration of Se). It is omitted, and not explained why only parameters relating to thyroid gland presented.

26. The introduction does not show the importance of the Se role, it focused rather on the thyroid gland, while the discussion relates only to Se. Moreover, Authors do not mention to any recommendation. Taking into account that Poland is close to Germany, and in Poland there is no given recommendation, Authors have to present German recommendation. It should be emphasing that among the Authors there is a German Author.

Round 2

Reviewer 2 Report

Comments to the Authors of manuscript number: animals-1779883R1  entitled “Selenium status and supplementation effects in pregnancy - study on mother-child pairs from a single-centre cohort”.

In general, Authors did not correct properly

1. L 70- if Authors inform in the title that the paper is about Se, the text should be consistent with it. If They want to discuss a potential interference Se effects with Iodine, the text and result should correspond to it.

2. L 102-103 This sentence does not have any sense. It means that someone who performed this procedure could take the sample with insufficient volume. Such situation cannot happen. Further, what does it mean: ”stored serum samples”.

3. L 109 – what is “optimal quality”? It should be explained

4. “11. L 117-125 – it is too descriptive and involves too many not important data like “centrifuged, sampled, frozen..” This chapter describes the Materials and Methods, which need to be descriptive and very exact, in order to enable other research groups to understand the methodology used in detail, and to allow replication of our study, which is the core of science. There is no space in the Materials and Methods section for non-descriptive information, as its major function is to describe the details of the study. However, the mentioned example has been modified according to your suggestion.”

Nutrients is the Journal at a very good reputability, and if someone writes the paper regularly to the Journal as such level, he/she knows that in Material and methods some activities can be omitted, because if it is rutinous sample preparing, and commonly known, there is no need to describe it in many details.

5. “12. In general Material and methods should be rephrased and prepared in manner which allow to easy follow the study design. The Materials and Methods section is to enable scientists in the field with similar education and professional status as the researchers to replicate the study. Here, two labs experiences in clinical research and analytical laboratory work have developed synergy during the study and during the manuscript preparation. All 13 authors consider the Materials and Methods section to be appropriate, in agreement with Reviewer 1.”

I do not have any doubts that Reviewer 1 has done the review in excellent manner, but the section of Material and methods should be described in such manner which allow others to replicate and check this study. The validation and threshold level of test used should be given. All abbreviations should be explained.

6. L 140 – “ according to experience” – it is not reason, but if there was preliminary clinical attempt, it is.  

7. Table 1 – where are units?

8. According to Table 1 the title of the table should be changed.

9. “20. L 427- “one serum sample per mother and per child only was available for analysis” what does it mean?
Again, it is proper English. To explain to reviewer, we meant than we took only one sample from mother and one from newborn. We consider this as a limitation, knowing that the second probe with material would enhance reliability of the results and provide the double-check, especially in newborns, although it was technically too difficult to obtain this amount of material from the cord blood.”

I did not ask Authors for English, but the meaning e.g. only one. It is known that cord blood collection is limited and this explanation should be given in the text.

10. Table 3  the first column is not readable.

11. “24. with the recommended daily supply of 60 μg for adult women, there is no need to increase the consumption of Se during pregnancy, while nursing women should take Se in the amount of 75 μg daily. It is not mentioned The supplement a women needs should be judged by its Se status. Women with high Se reserves and high expression of selenoproteins may not be in need of any supplement at all. It depends on their dietary patterns and areas of residency. However, when living in Poland and e.g. avoiding fish, eggs, milk and meat, higher supplemental amounts of Se are needed in order to enable sufficient selenoprotein expression. This is a most personalized issue, and this is exactly what we discuss in the discussion section.

From this point of view, the diet should be presented.

12. I agree that the nationality of Authors do not matter, but this study involves Authors from two important countries: from Poland, where there is no recommendation, and German where it is. When the paper is written in co-operation such information are clear. The nationality of reviewer does not matter. Especially, that the recommendation of medical societies (the group from Endocrinological Clinic in Warsaw, Poland) speaks  about the use of the German recommendation.  It should be clarified.

Kipp AP, Strohm D, Brigelius-Flohé R et al.: German Nutrition Society (DGE): Revised reference values for selenium intake. J Trace Elem Med Biol 2015; 32: 195-199.

I am so sorry, if Authors understood this comment opposite.
